# On Epistemics in Expected Free Energy for Linear Gaussian State Space Models

**DOI:** 10.3390/e23121565

**Published:** 2021-11-24

**Authors:** Magnus T. Koudahl, Wouter M. Kouw, Bert de Vries

**Affiliations:** 1Department of Electrical Engineering, Eindhoven University of Technology, 5612 AZ Eindhoven, The Netherlands; w.m.kouw@tue.nl (W.M.K.); Bert.de.Vries@tue.nl (B.d.V.); 2GN Hearing, JF Kennedylaan 2, 5612 AB Eindhoven, The Netherlands

**Keywords:** active inference, epistemics, expected free energy, free energy principle, linear Gaussian dynamical system

## Abstract

Active Inference (AIF) is a framework that can be used both to describe information processing in naturally intelligent systems, such as the human brain, and to design synthetic intelligent systems (agents). In this paper we show that Expected Free Energy (EFE) minimisation, a core feature of the framework, does not lead to purposeful explorative behaviour in linear Gaussian dynamical systems. We provide a simple proof that, due to the specific construction used for the EFE, the terms responsible for the exploratory (epistemic) drive become constant in the case of linear Gaussian systems. This renders AIF equivalent to KL control. From a theoretical point of view this is an interesting result since it is generally assumed that EFE minimisation will always introduce an exploratory drive in AIF agents. While the full EFE objective does not lead to exploration in linear Gaussian dynamical systems, the principles of its construction can still be used to design objectives that include an epistemic drive. We provide an in-depth analysis of the mechanics behind the epistemic drive of AIF agents and show how to design objectives for linear Gaussian dynamical systems that do include an epistemic drive. Concretely, we show that focusing solely on epistemics and dispensing with goal-directed terms leads to a form of maximum entropy exploration that is heavily dependent on the type of control signals driving the system. Additive controls do not permit such exploration. From a practical point of view this is an important result since linear Gaussian dynamical systems with additive controls are an extensively used model class, encompassing for instance Linear Quadratic Gaussian controllers. On the other hand, linear Gaussian dynamical systems driven by multiplicative controls such as switching transition matrices do permit an exploratory drive.

## 1. Introduction

Active Inference (AIF) is a mathematical description of information processing in intelligent systems. In brief it states that agents, originally biological but in later years also synthetic, act to minimise their surprise by seeking out stimuli and states that are compatible with their model of the world. AIF is an attractive framework for designing synthetic agents since AIF agents possess a well-balanced drive towards both explorative (epistemic) and exploitative (pragmatic, goal-driven) behaviour. These characteristics follow from choosing the Expected Free Energy (EFE) as the objective function for planning.

In this paper we explicitly derive the equations for applying AIF in linear Gaussian dynamical systems (LGDS) with the EFE objective. In doing so we uncover a novel result showing that, in the case of linear models, the epistemic term of the EFE objective becomes constant. This means that any application of EFE in LGDS will not lead to exploration and the resulting agents will engage in purely goal-driven behaviour. The proof is given in Section 5.5. The remainder of the paper is structured as an in-depth analysis of the AIF framework and the mechanisms driving its claims to epistemic behaviour. We isolate the epistemic term of the EFE and identify it as a (bound on) mutual information (MI). We then show that, when considering epistemics in isolation instead of the full EFE construct, it is still possible to generate an epistemic drive using the machinery of AIF. Isolating epistemics corresponds to a special case of EFE where priors on future observations are left unspecified [1,2]. We analyze the behaviour of the resulting epistemic drive and show that, for the case of additive controls, the epistemic drive is independent of state transitions and only depends on the prior variance associated with the belief over the control signal. On the other hand, LGDS driven by multiplicative control signals do exhibit a dependence between state transitions and the epistemic drive. Prior work on AIF in LGDS such as [3,4,5,6,7], have focused mostly on the goal-directed components of the AIF framework. The results, while impressive, largely do not address questions of epistemics and exploration. This means that in cases where AIF is applied to LGDS, EFE and the resulting desirable exploratory drive have so far not been thoroughly investigated. Our results show that, provided the model in question can be cast as a LGDS, incorporating EFE does not lead to meaningful exploration. The present paper makes the following contributions:We derive the filtering and planning equations for AIF using EFE in LGDS, Section 4 and Section 5.We consider the epistemic term of EFE in isolation and show that in the case of additive controls actions become decoupled from state transitions when computing the epistemic term of EFE, Section 5.3. Therefore, we do not find meaningful exploration in this case.We show that in the case of multiplicative controls, meaningful exploratory behaviour re-emerges when isolating the epistemic term of EFE, Section 5.4.We prove that when considering the full EFE construct, parts of the instrumental and epistemic value terms cancel each other out. This renders the epistemic value constant. In turn, the EFE functional becomes equivalent to KL control plus an additive constant, Section 5.5.Finally, we provide simulations that corroborate our claims. We first demonstrate the differences in exploration when considering purely epistemic agents using both additive and multiplicative control signals. Finally we show that LGDS agents optimising the full EFE do not exhibit epistemic drives under any circumstances, Section 6.

The core message is thus that translating AIF to the linear Gaussian case presents unique challenges, specifically because the exploration/exploitation trade-off that follows from EFE minimisation does not manifest. Code to reproduce our experiments is available at github.com/biaslab/efe_lgds (accessed on 19 September 2021).

## 2. Exploration and Exploitation

In this section we aim to introduce the concepts of exploration and exploitation on intuitive grounds before commencing with our formal analysis. Exploitation refers to goal directed behaviour. An agent that engages in exploitation performs actions that are aimed at optimising some measure of preferences which we will refer to as “Instrumental value”. As an example, consider minimisation of mean squared error, cross entropy or a similar cost function. Exploration on the other hand, refers to behaviour directed at collecting information about the environment in which the agent is embedded. An agent that engages in exploration performs actions that are aimed at acquiring further information about its environment. We will refer to any metric that quantifies the value of gathering information as “Epistemic value”. Optimising epistemic value biases the agent towards actions that gather information. We will refer to this bias in action selection as an “Epistemic drive”. There are many candidates for the epistemic value term. We will briefly consider two that are particularly relevant for the present analysis. This will not be a formal comparison but an intuitive introduction to the qualitative differences in behaviour that can be expected from agents that employ different epistemic value terms. First, we can consider agents that aim to maximise entropy (uncertainty). For such an agent, the epistemic drive biases it towards seeking out areas of state space where uncertainty is high. By repeatedly visiting uncertain areas of state space, the agent collects observations in said areas which in turn reduces uncertainty. As an example, we can consider an agent trying to navigate an arena. The agent is equipped with a sensor and the arena is subject to strong winds that induce sensor noise by pushing the agent around. In this case maximising entropy drives the agent to seek out parts of the arena where the winds (and corresponding sensor noise) are high. This means the agent collects information primarily in areas where more observations are needed, due to increased sensor noise. Second, we can consider an agent that aims to maximise MI, also known as Information Gain. We provide a formal treatment of MI in Section A.4. Intuitively, MI scores the reduction in uncertainty that the agent expects given a particular observation. In the present example, an agent that optimises MI might correctly identify that although windy areas are noisy, collecting information in those areas is unlikely to reduce uncertainty because the winds will remain high. Instead the agent will prefer to move towards areas that have less wind, in order to obtain more accurate measurements. This is the approach taken by AIF agents when optimising EFE [8,9]. Optimising both instrumental and epistemic value terms by selecting actions necessarily entail a trade-off between short term gains (exploitation) and gathering information in order to perform better in the future (exploration). Having agents that are able to optimally balance this trade-off is therefore desirable because it allows for autonomous systems that are able to learn to navigate novel environments in order to achieve desired goals. A core feature of the EFE is that it presents a single objective functional that encompasses both instrumental and epistemic value terms [8,9]. In order to formally unpack how AIF manifests both instrumental and epistemic value terms, we now need to detail the LGDS model that specifies our agent before deriving the equations for computing the EFE objective.

## 3. Generative Model

AIF is fundamentally a model-based approach [8,10]. As such, the core part of an agent is given by a generative model. Given a generative model, the agent engages in a perception-action loop with its environment. In practice this means the agent will, at any time step, absorb a new observation and emit a new action. The first step is always perception, followed by action selection and emission. Letting $x\in R^{d}$ denote observations, $z\in R^{n}$ a latent state vector and *u* actions (we will use “actions” and “controls” interchangeably to refer to *u* throughout), the generative model for an agent at a single time step, indicated by subscripts *t*, has the form
(1)$$\begin{array}{cc} p(x_{t},z_{t}|u_{t},z_{t-1}) & =\underset{\mathrm{Likelihood}}{\underset{︸}{p(x_{t}|z_{t})}}\underset{\mathrm{State}\;\mathrm{transition}}{\underset{︸}{p(z_{t}|z_{t-1},u_{t})}}\;. \end{array}$$

This form can be extended, for example by including parameters θ. If applied recursively, this model corresponds to a discrete-time state space model. A common approach when designing AIF agents is to work directly with a policy defined as a *particular* sequence of actions $u_{t+1:T}$ [8,11,12,13] where the subscript denotes discrete time steps ranging from the next time step t+1 to some known planning horizon *T*. In (Equation 1) we indicate this by explicitly conditioning on $u_{t}$. This sequence of actions is then considered either as an explicit vector of control signals [8,13,14,15] or amortised for instance by neural networks [16,17,18,19]. Proceeding in this way leads to a particular scheme for action selection which we will detail in Section 5.

In this paper we consider the case of LGDS with multiplicative or additive controls. To clarify the distinction between additive and multiplicative controls, we define “multiplicative controls” as state transitions of the form
(2)$$\begin{array}{c} p\left(z_{t}|z_{t-1},u_{t}\right)=N\left(z_{t}|B\left(u_{t}\right)z_{t-1},\Sigma_{z}\right)\;, \end{array}$$
where $z_{t}\in R^{n}$ is a latent state vector, $u_{t}$ is a discrete control signal and $B(u_{t})$ is the transition matrix. We consider the case where the control signal functions as a selector variable. Formally we define a vector of candidate transition matrices $\left[B_{1},B_{2},\ldots B_{S}\right]$ and let
(3)$$\begin{array}{cc} B(u_{t}) & =\prod_{s=1}^{S}B_{s}^{u_{ts}}\;. \end{array}$$Here $u_{t}$ is a one-hot encoded vector $u_{t}=\left[u_{t1},\ldots ,u_{tS}\right]$ that takes values in $u_{ts}\in \left\{0,1\right\}$ and where $\sum_{s=1}^{S}u_{ts}=1$. Each $B_{s}$ is raised to the power given by $u_{ts}$, which means that only the selected $B_{s}$ will be active. The control signal therefore influences state transitions by selecting the transition matrix *B* directly.

We can visualise this model using the Forney-style Factor Graph (FFG) formalism [20]. In an FFG, each edge represents a variable and each node a factor. An edge is connected to a node if and only if the corresponding variable is an argument of that factor. Each edge connects at most two nodes. When a variable is an argument of more than two factors, we can circumvent the two node per edge limit by linking edges together through equality factors. This effectively creates an auxiliary variable (a new edge) for which the posterior beliefs are constrained to be equal to the beliefs for the original variable. The new edge can also be attached to two factors, so by adding equality factors we can use the same variable as an argument of multiple factors. Observed variables and clamped parameters are denoted by a small black square and selected actions by a small black diamond. The selection mechanism described by (Equation 3) is denoted by the multiplexer (MUX) node. Instead of cluttering the graph by drawing the full set of $\left[B_{1},B_{2},\cdots B_{S}\right]$ candidate transition matrices as separate nodes, we denote them by a shaded circle. The circle contains *S* nodes and their corresponding outgoing edges all connect to the MUX node. For a further introduction to FFGs, see [21,22]. The FFG of the multiplicative model can be seen in Figure 1.

For comparison, we now consider the case of additive controls. For the additive case the generative model is again given by (Equation 1). However exact model specification is a little more involved. We consider transition models of the form
(4)$$\begin{array}{c} p\left(z_{t}|z_{t-1},u_{t}\right)=N\left(z_{t}|Bz_{t-1}+b\left(u_{t}\right),\Sigma_{z}\right)\;, \end{array}$$
where $B\in R^{n\times n}$ is a known transition matrix and $b\left(u_{t}\right)\in R^{n}$ is a vector function that adds to the latent state. To rigorously compare the multiplicative and additive control cases, $u_{t}$ must remain a categorical selector variable. To that end, we introduce an auxiliary variable $b_{t}$. The purpose of $b_{t}$ is to allow $u_{t}$ to function as a categorical selector variable. Instead of selecting between transition matrices $B_{s}$, $u_{t}$ now selects the parameters $\Theta_{s}=\left\{\mu_{s},\Sigma_{s}\right\}$ of a Gaussian input signal. Formally, we will write the generative model as
(5a)$$\begin{array}{cc} p(z_{t}|z_{t-1},u_{t}) & =\int p\left(z_{t}|z_{t-1},b_{t}\right)p\left(b_{t}|u_{t}\right)db_{t} \end{array}$$
(5b)$$\begin{array}{cc} \; & =\int N\left(z_{t}|Bz_{t-1}+b_{t},\Sigma_{z}\right)\prod_{s=1}^{S}N{\left(b_{t}|\Theta_{s}\right)}^{u_{ts}}db_{t}\;. \end{array}$$
where we recognise a similar selection mechanism of (Equation 3) in the second term of ([Disp-formula FD5a-entropy-23-01565]). Selecting an action means fixing $u_{t}={\hat{u}}_{t}$ which leads to selecting one of the candidate Gaussian distributions. With only a single Gaussian surviving, integration over $b_{t}$ becomes straightforward and yields
(6)$$\begin{array}{cc} p(z_{t}|z_{t-1},{\hat{u}}_{t}) & =N(z_{t}|Bz_{t-1}+{\hat{\mu }}_{u_{t}},\Sigma_{z}+{\hat{\Sigma }}_{u_{t}})\;, \end{array}$$
where ${\hat{\Theta }}_{s}=\left\{{\hat{\mu }}_{u_{t}},{\hat{\Sigma }}_{u_{t}}\right\}$ represent the parameters of the Gaussian distribution selected by ${\hat{u}}_{t}$.

The factor graph of the additive model is shown in Figure 2 where the MUX node now selects between $\Theta_{1:s}$. In both the multiplicative and additive settings, we employ a likelihood term of the form:(7)$$\begin{array}{cc} p(x_{t}|z_{t}) & =N(x_{t}|Az_{t},\Sigma_{x})\;, \end{array}$$
where $A\in R^{d\times n}$ is a known emission matrix and $\Sigma_{x}$ represents measurement or observation noise.

Having established the relevant model structures, we now examine the perception/action loop starting with perception.

## 4. Perception as Bayesian Filtering

The perception part of the action/perception loop involves making inference about observed data and can be cast as a Bayesian filtering problem. This part of the process describes the agent inferring the hidden state of its environment based on the sequence of actions taken so far, the resulting sequence of states visited and the accompanying observations. We can write the resulting inference problem in an intuitive way as a prediction-correction process:(8)$$\begin{array}{c} \underset{\mathrm{posterior}}{\underset{︸}{p(z_{t}|x_{1:t})}}=\frac{p(x_{t}|z_{t})}{\underset{\begin{array}{c} \mathrm{correction}\\ \mathrm{based}\;\mathrm{on}\;x_{t} \end{array}}{\underset{︸}{p(x_{t}|x_{1:t-1})}}}\times \underset{\begin{array}{c} \mathrm{prediction}\;\mathrm{of}\;z_{t}\\ \mathrm{based}\;\mathrm{on}\;x_{1:t-1} \end{array}}{\underset{︸}{p(z_{t}|x_{1:t-1})}}\;. \end{array}$$ The above shows how the inference over states can be accomplished recursively (due to the model obeying the Markov property) by first computing a prediction for the next hidden state $z_{t}$ to generate a prior belief which is then updated in a correction step based on the observed data point $x_{t}$.

The Bayesian filtering problem is generic. To see how it translates to our case, we can expand the prior predictive in terms of our generative model
(9)$$\begin{array}{cc} \underset{\mathrm{state}\;\mathrm{posterior}}{\underset{︸}{p(z_{t}|x_{1:t})}} & =\frac{\overset{\mathrm{likelihood}}{\overset{︷}{p(x_{t}|z_{t})}}}{\underset{\mathrm{evidence}}{\underset{︸}{p(x_{t}|x_{1:t-1})}}}\overset{\mathrm{prior}\;\mathrm{predictive}\;p(z_{t}|x_{1:t-1})}{\overset{︷}{\;\int \int \;\underset{\mathrm{state}\;\mathrm{transition}}{\underset{︸}{p(z_{t}|z_{t-1},u_{t})}}\underset{\begin{array}{c} \mathrm{control}\\ \mathrm{signal} \end{array}}{\underset{︸}{\delta (u_{t}-{\hat{u}}_{t})}}\underset{\mathrm{state}\;\mathrm{prior}}{\underset{︸}{p(z_{t-1}|x_{1:t-1})}}dz_{t-1}du_{t}}}\;, \end{array}$$We use $\delta (u_{t}-{\hat{u}}_{t})$ where δ is the Dirac-δ to fix the value of $u_{t}$ to the selected action ${\hat{u}}_{t}$ for that particular time step. The particular value chosen for ${\hat{u}}_{t}$ is the result of the action selection procedure described in Section 5. The evidence term is given by
(10)$$\begin{array}{cc} p(x_{t}|x_{1:t-1}) & =\int p\left(x_{t}|z_{t}\right)\left(\;\int \int \;p\left(z_{t}|z_{t-1},u_{t}\right)\delta \left(u_{t}-{\hat{u}}_{t}\right)p\left(z_{t-1}|x_{1:t-1}\right)dz_{t-1}du_{t}\right)dz_{t}\;. \end{array}$$For the LGDS models considered in this paper, filtering can be performed using the Kalman filtering equations. We will first work this out explicitly in the multiplicative case and then in the additive. To show how to perform filtering in the multiplicative model, we start by assuming that the agent has selected an action $u_{t}={\hat{u}}_{t}$ by the procedure described in Section 5. We can then calculate the prior predictive distribution of (Equation 9) according to our model specification (Equation 1) as
(11a)$$\begin{array}{cc} \;\;\;p(z_{t}|x_{1:t-1})\; & =\;\int \int \;p\left(z_{t}|z_{t-1},u_{t}\right)\delta \left(u_{t}-{\hat{u}}_{t}\right)p\left(z_{t-1}|x_{1:t-1}\right)dz_{t-1}du_{t} \end{array}$$
(11b)$$\begin{array}{cc} \; & \;\;\;\;\;\;\;\;\;\;\;\;=\;\int \int \;\underset{\mathrm{State}\;\mathrm{transition}}{\underset{︸}{N(z_{t}|B\left(u_{t}\right)z_{t-1},\Sigma_{z})}}\underset{\begin{array}{c} \mathrm{Selected}\\ \mathrm{control}\;\mathrm{signal} \end{array}}{\underset{︸}{\delta (u_{t}-{\hat{u}}_{t})}}\underset{\mathrm{State}\;\mathrm{prior}}{\underset{︸}{N(z_{t-1}|\mu_{z_{t-1}},\Sigma_{z_{t-1}})}}dz_{t-1}du_{t} \end{array}$$
(11c)$$\begin{array}{cc} \; & \;\;\;\;\;\;=\int N\left(z_{t}|B\left({\hat{u}}_{t}\right)z_{t-1},\Sigma_{z}\right)N\left(z_{t-1}|\mu_{z_{t-1}},\Sigma_{z_{t-1}}\right)dz_{t-1} \end{array}$$
(11d)$$\begin{array}{cc} \; & =N(z_{t}|\underset{\mu_{z_{t}}^{-}}{\underset{︸}{{\hat{B}}_{t}\mu_{z_{t-1}}}},\underset{\Sigma_{z_{t}}^{-}}{\underset{︸}{{\hat{B}}_{t}\Sigma_{z_{t-1}}{\hat{B}}_{t}^{T}+\Sigma_{z}}})\;, \end{array}$$
which we recognise as the prediction step of a Kalman filter [23]. We use the superscript ${}^{-}$ notation to indicate that the variable in question is not based on the full data set $x_{1:t}$ but instead on a smaller data set $x_{1:t-1}$. In moving from ([Disp-formula FD11b-entropy-23-01565]) to ([Disp-formula FD11c-entropy-23-01565]) we rely on the sifting property of the Dirac-δ to substitute the selected value for $u_{t}$ in (Equation 3). Since $B(u_{t})$ is a function of $u_{t}$ and $u_{t}$ is now fixed to ${\hat{u}}_{t}$, we can directly substitute the selected parameterisation by setting $B\left(u_{t}\right)={\hat{B}}_{t}$ where ${\hat{B}}_{t}$ denotes the parameterisation given by the selected $B_{s}$. This takes us from ([Disp-formula FD11b-entropy-23-01565]) to ([Disp-formula FD11c-entropy-23-01565]). Finally we can rely on standard results for linearly related and jointly Gaussian variables to go from ([Disp-formula FD11c-entropy-23-01565]) to ([Disp-formula FD11d-entropy-23-01565]), see for example [23] [Section A.1] for details of this move in the context of Gaussian state space models or Section A.2 for an abbreviated version. For the additive control case, we can calculate the prior predictive distribution in a similar fashion. Starting from the model definition (Equation 1), we can write
(12a)$$\begin{array}{cc} \;\;\;p(z_{t}|x_{1:t-1})\; & =\;\int \int \int \;p\left(z_{t}|z_{t-1},b_{t}\right)p\left(b_{t}|u_{t}\right)\delta \left(u_{t}-{\hat{u}}_{t}\right)p\left(z_{t-1}|x_{1:t-1}\right)du_{t}db_{t}dz_{t-1} \end{array}$$
(12b)$$\begin{array}{cc} \; & \;\;\;\;\;\;\;\;\;\;\;\;\;\;\;\;\;\;=\;\int \int \int \;\;\underset{\mathrm{State}\;\mathrm{transition}}{\underset{︸}{N\left(z_{t}|Bz_{t-1}\;+\;b_{t},\Sigma_{z}\right)\prod_{s=1}^{S}N{\left(b_{t}|\mu_{s},\Sigma_{s}\right)}^{u_{ts}}}}\;\underset{\begin{array}{c} \mathrm{Selected}\\ \mathrm{control}\;\mathrm{signal} \end{array}}{\underset{︸}{\delta (u_{t}\;-\;{\hat{u}}_{t})}}\underset{\mathrm{State}\;\mathrm{prior}}{\underset{︸}{N(z_{t-1}|\mu_{z_{t-1}},\Sigma_{z_{t-1}})}}du_{t}db_{t}dz_{t-1} \end{array}$$
(12c)$$\begin{array}{cc} \; & \;\;\;\;\;\;\;\;\;\;\;=\;\int \int \;N\left(z_{t}|Bz_{t-1}+b_{t},\Sigma_{z}\right)\prod_{s=1}^{S}N{\left(b_{t}|\mu_{s},\Sigma_{s}\right)}^{{\hat{u}}_{ts}}N\left(z_{t-1}|\mu_{z_{t-1}},\Sigma_{z_{t-1}}\right)db_{t}dz_{t-1} \end{array}$$
(12d)$$\begin{array}{cc} \; & \;\;\;\;\;\;\;\;\;\;=\;\int \int \;N\left(z_{t}|Bz_{t-1}+b_{t},\Sigma_{z}\right)\underset{\begin{array}{c} \mathrm{Selected}\\ \;\mathrm{parameters},\;{\hat{\Theta }}_{s} \end{array}}{\underset{︸}{N(b_{t}|{\hat{\mu }}_{u_{t}},{\hat{\Sigma }}_{u_{t}}}})N\left(z_{t-1}|\mu_{z_{t-1}},\Sigma_{z_{t-1}}\right)db_{t}dz_{t-1} \end{array}$$
(12e)$$\begin{array}{cc} \; & \;\;\;\;\;=\int N\left(z_{t}|Bz_{t-1}+{\hat{\mu }}_{u_{t}},\Sigma_{z}+{\hat{\Sigma }}_{u_{t}}\right)N\left(z_{t-1}|\mu_{z_{t-1}},\Sigma_{z_{t-1}}\right)dz_{t-1} \end{array}$$
(12f)$$\begin{array}{cc} \; & =N(z_{t}|\underset{\mu_{z_{t}}^{-}}{\underset{︸}{B\mu_{z_{t-1}}+{\hat{\mu }}_{u_{t}}}},\underset{\Sigma_{z_{t}}^{-}}{\underset{︸}{\Sigma_{z}+{\hat{\Sigma }}_{u_{t}}+B\Sigma_{z_{t-1}}B^{T}}})\;. \end{array}$$We again denote the selected parameters for the additive control at the *k*-th time step as ${\hat{\mu }}_{u_{k}}$ and ${\hat{\Sigma }}_{u_{k}}$. To proceed from ([Disp-formula FD12b-entropy-23-01565]) to ([Disp-formula FD12c-entropy-23-01565]) we rely on the sifting property of the Dirac-δ and substitute the selected value for $u_{t}$. The move from ([Disp-formula FD12c-entropy-23-01565]) to ([Disp-formula FD12d-entropy-23-01565]) acknowledges that only the selected parameterisation is active once we substitute $u_{t}={\hat{u}}_{t}$ as covered in Section 3. The final steps from ([Disp-formula FD12d-entropy-23-01565]) to ([Disp-formula FD12e-entropy-23-01565]) and from ([Disp-formula FD12e-entropy-23-01565]) to ([Disp-formula FD12f-entropy-23-01565]) uses standard results for multiplication and marginalisation of jointly Gaussian variables for which we again refer to [23] [Section A.1] and Section A.2. In summary, we see that when $u_{t}={\hat{u}}_{t}$ the model specification given in ([Disp-formula FD5b-entropy-23-01565]) reduces to a standard LGDS and can be updated using the prediction step of the Kalman filtering equations.

Both the additive and multiplicative models use similar likelihood models, meaning they can be updated using the same Kalman correction step. To perform the Kalman correction step, we need to apply Bayes rule
(13)$$\underset{\mathrm{posterior}}{\underset{︸}{p(z_{t}|x_{1:t})}}\underset{\mathrm{evidence}}{\underset{︸}{p(x_{t}|x_{1:t-1})}}=\underset{\mathrm{likelihood}}{\underset{︸}{p(x_{t}|z_{t})}}\underset{\mathrm{prior}}{\underset{︸}{p(z_{t}|x_{1:t-1})}}\;,$$
where the factor on the right-hand site (RHS) are given and the terms on the left-hand side are the desired factors. This equation can be solved analytically. First, we evaluate the RHS as
(14a)$$\begin{array}{cc} p\left(x_{t}|z_{t}\right)p\left(z_{t}|x_{1:t-1}\right)\; & =N\left(x_{t}|Az_{t},\Sigma_{x}\right)N\left(z_{t}|\mu_{z_{t}}^{-},\Sigma_{z_{t}}^{-}\right) \end{array}$$
(14b)$$\begin{array}{cc} \; & \;\;\;\;\;\;\;\;\;\;\;\;=N\left(\left[\begin{array}{c} z_{t}\\ x_{t} \end{array}\right]|\left[\begin{array}{c} \mu_{z_{t}}^{-}\\ A\mu_{z_{t}}^{-} \end{array}\right],\left[\begin{array}{cc} \Sigma_{z_{t}}^{-} & \Sigma_{z_{t}}^{-}A^{T}\\ A\Sigma_{z_{t}}^{-} & A\Sigma_{z_{t}}^{-}A^{T}+\Sigma_{x} \end{array}\right]\right)\;. \end{array}$$Then if, for notational convenience, we rewrite the covariance matrix as
(15)$$\begin{array}{c} \left[\begin{array}{cc} \Sigma_{11} & \Sigma_{12}\\ \Sigma_{21} & \Sigma_{22} \end{array}\right]≜\left[\begin{array}{cc} \Sigma_{z_{t}}^{-} & \Sigma_{z_{t}}^{-}A^{T}\\ A\Sigma_{z_{t}}^{-} & A\Sigma_{z_{t}}^{-}A^{T}+\Sigma_{x} \end{array}\right]\;, \end{array}$$([Disp-formula FD14b-entropy-23-01565]) can be written as the product of the state posterior
(16)$$\begin{array}{cc} p(z_{t}|x_{1:t})\; & =N(z_{t}|\mu_{z_{t}}^{-}+\Sigma_{12}\Sigma_{22}^{-1}\left(x_{t}-A\mu_{z_{t}}^{-}\right),\Sigma_{11}-\Sigma_{12}\Sigma_{22}^{-1}\Sigma_{21})\;, \end{array}$$
and evidence
(17)$$\begin{array}{cc} p(x_{t}|x_{1:t-1}) & =N(x_{t}|A\mu_{z_{t}}^{-},\Sigma_{22})\;. \end{array}$$
due to the theorem for decomposing a multi-variate Gaussian into the product of a conditional distribution [23] [Appendix A.1].

Finally, we can also calculate the conditional distribution $p(x_{t}|z_{t},x_{1:t-1})$ [23] [Section A.1]. While this is not required for solving the Bayesian filtering problem, it will prove useful for deriving the epistemic value term (as derived in Section A.4) used for the action selection procedure described in Section 5. We can find it as
(18)$$\begin{array}{cc} p(x_{t}|z_{t},x_{1:t-1})\; & =N(x_{t}|A\mu_{z_{t}}^{-}+\Sigma_{21}\Sigma_{11}^{-1}\left(z_{t}-\mu_{z_{t}}^{-}\right),\Sigma_{22}-\Sigma_{21}\Sigma_{11}^{-1}\Sigma_{12})\;. \end{array}$$

Having described perception as Bayesian filtering, we now turn our attention to action selection under AIF.

## 5. Action Selection under Active Inference

When we are interested in constructing AIF agents, arguably the core task is that of action selection. Under AIF we solve this task by first computing a prior over future control signals. Technically, we seek to compute
(19)$$\begin{array}{c} p\left(u_{t+1:T}\right)\propto \exp\left(-G\left(u_{t+1:T}\right)\right)\;, \end{array}$$
i.e., the prior on controls is a softmax function of $G(u_{t+1:T})$ [8] [Equation (7)]. Here, $G(u_{t+1:T})$ denotes the expected free energy (EFE) for a policy that extends into the future until a known horizon *T*. We further discuss the computation of $G(u_{t+1:T})$ in Section 5.1. To obtain the control prior at time t+1, we can marginalise this distribution as
(20a)$$\begin{array}{cc} p(u_{t+1})\; & =\int \cdots \int p\left(u_{t+1:T}\right)du_{t+2}\cdots du_{T} \end{array}$$
(20b)$$\begin{array}{cc} \; & \;\;\;\;\;\;\propto \int \cdots \int \exp\left(-G\left(u_{t+1:T}\right)\right)du_{t+2}\cdots du_{T}\;. \end{array}$$If we assume independent control priors for each time step, that is, if we assume $p\left(u_{t+1:T}\right)=\prod_{k=t+1}^{T}p\left(u_{k}\right)$, or equivalently,
(21)$$\begin{array}{cc} \exp(-G\left(u_{t+1:T}\right)) & =\prod_{k=t+1}^{T}\exp\left(-G\left(u_{k}\right)\right)\;, \end{array}$$
then the marginalisation (20) evaluates to
(22)$$\begin{array}{cc} p(u_{t+1}) & \propto \exp\left(-G(u_{t+1})\right)\;. \end{array}$$Since the marginalisation procedure is identical for any other time step, we can deduce that the total EFE for a policy is equal to the exponentiated sum of EFEs at individual time steps. That is
(23)$$\begin{array}{cc} p(u_{t+1:T}) & \propto \exp\left(-G(u_{t+1:T})\right)=\prod_{k=t+1}^{T}\exp\left(-G\left(u_{k}\right)\right)=\exp\left(-\sum_{k=t+1}^{T}G\left(u_{k}\right)\right)\;. \end{array}$$This suggests a recursive scheme over time steps for computing policy priors, similar to the proposal by [13]. In the AIF literature the executed action is then commonly sampled from $p(u_{t+1:T})$ and emitted to the environment [8,13]. Other action selection approaches such as selecting the MAP estimate (the argmax or mode of the distribution) are also possible. We now turn our attention to how the expected free energy is computed.

### 5.1. Computing G—Expected Free Energy

The EFE is an AIF specific construct that attempts to model what the variational free energy would be at a future time step, conditioned on a particular sequence of actions. Of special interest is the decomposition of EFE into an epistemic (explorative) term and an instrumental (exploitative) term. It is due to this decomposition that AIF claims an adaptive trade-off between exploration and exploitation [8,13]. Note that we provide the derivation here only for the simplest case. There are extensions to the EFE such as [24] that include additional terms which induce changes in the agent’s behaviour. Including these additional terms are not necessary for our core argument so we omit them here and refer interested readers to [24,25]. To show how we arrive at our formulation for EFE, we first need to introduce variational inference. While the filtering equations in Section 4 permit analytical solutions by applying Bayes rule directly, this is often not the case as solving the required integrals can become intractable. In those cases, we can instead approximate the exact solution $p(z_{t}|x_{1:t})$ by a recognition density $q(z_{t})$. Formally we accomplish this by minimising the KL divergence between the exact solution and our recognition density
(24)$$\begin{array}{cc} KL[q\left(z_{t}\right)||p\left(z_{t}|x_{1:t}\right)] & =\int q\left(z_{t}\right)\log\frac{q(z_{t})}{p(z_{t}|x_{1:t})}dz_{t} \end{array}$$

If we now multiply and divide by $p(x_{t}|x_{1:t-1})$ inside the log-operator,
(25)$$\begin{array}{cc} KL[q\left(z_{t}\right)||p\left(z_{t}|x_{1:t}\right)]\; & =\int q\left(z_{t}\right)\left[\log\frac{q(z_{t})}{p(x_{t},z_{t}|x_{1:t-1})}+\log p\left(x_{t}|x_{1:t-1}\right)\right]dz_{t} \end{array}$$
(26)$$\begin{array}{cc} \; & \;\;\;\;\;\;\;=\underset{\mathrm{VFE}\;F[q]}{\underset{︸}{\int q\left(z_{t}\right)\log\frac{q(z_{t})}{p(x_{t},z_{t}|x_{1:t-1})}dz_{t}}}+\underset{\text{log-evidence}}{\underset{︸}{\log p(x_{t}|x_{1:t-1})}}\;. \end{array}$$We obtain the variational free energy (VFE) F[q] by noting that $\log p(x_{t}|x_{1:t-1})$ is not dependent on $z_{t}$. Since the term $p(x_{t},z_{t}|x_{1:t-1})$ in the denominator of F[q] is given by our generative model, we can choose constraints on $q(z_{t})$ to make optimization of (Equation 26) tractable. Minimizing F[q] then constitutes an upper bound on $-\log p(x_{t}|x_{1:t-1})$, meaning we can optimise (Equation 26) to obtain an approximate solution to our original inference problem. We define the optimal recognition density $q^{*}$ as the one that minimises F[q]:(27)$$\begin{array}{c} q^{*}=\underset{q}{\arg\;\min}\;\;F\left[q\right] \end{array}$$For further background on variational inference we refer interested readers to the seminal works by [26,27]. Now we are ready to introduce the EFE. To do so, we start by obtaining our best estimate of the time step *k* in question by integrating out contributions from past time steps as
(28)$$\begin{array}{cc} p(x_{k},z_{k}|u_{k}) & =\int p\left(x_{k},z_{k}|z_{k-1},u_{k}\right)p\left(z_{k-1}|x_{1:t}\right)dz_{k-1}\;, \end{array}$$
where we write *k* instead of *t* to indicate that we are referring to an arbitrary time step within the planning horizon t<k≤T. $p(z_{k-1}|x_{1:t})$ denotes the posterior state estimate at the previous time step given all available observations. For notational brevity we suppress the dependency on $x_{1:t}$ on the LHS. Unless otherwise noted, all distributions are conditioned on prior observations moving forward. For LGDS $p(z_{k-1}|x_{1:t})$ is available from recursive application of the filtering equations described in Section 4. For k=t+1 it is given by (Equation 16) and for k>t+1, $p(z_{k-1}|x_{1:t})$ is given by (11) in the case of multiplicative controls and (12) in the case of additive controls. When $p(z_{k-1}|x_{1:t})$ can not be obtained through application of Bayes rule (providing the exact solution), one can employ variational inference (providing an approximate solution). In that case derivations must instead proceed in terms of the approximate posterior $q(z_{k-1}|x_{1:t})$. Now we can write out the variational free energy conditioned on a particular action $u_{k}={\hat{u}}_{k}$ and recognition density as $F[q;u_{k}]$. Note that while F[q] is a *functional* (a function of a function) of *q*, we also explicitly include conditioning on action given by the *parameter*$u_{k}$. To differentiate, we separate them with a semicolon when writing $F[q;u_{k}]$. Given the factorisation in (Equation 23), it is sufficient to consider a single time step *k* since we can substitute any value for *k*. This gives us
(29)$$\begin{array}{cc} F[q;u_{k}] & =\int q\left(z_{k}|u_{k}\right)\log\frac{q(z_{k}|u_{k})}{p(x_{k},z_{k}|u_{k})}dz_{k}\;. \end{array}$$However this expression includes observations $x_{k}$ which are not available, since we are working with time steps in the future (t<k≤T) and the future is by definition not observed yet. To alleviate this issue, we can take the expectation of this expression with respect to the data generating distribution over observations. When the data generating distribution is available from the generative model, we can equivalently write $p(x_{k}|z_{k})$ instead of $q(x_{k}|z_{k})$. This gives the expression for the expected free energy at the *k*’th time step:(30)$$\begin{array}{cc} G[q;u_{k}] & =\underset{\mathrm{Expected}\;F[q;u_{k}]\;\sin\mathrm{ce}\;x_{k}\;\mathrm{is}\;\mathrm{not}\;\mathrm{yet}\;\mathrm{observed}}{\underset{︸}{\;\int \int \;q\left(x_{k}|z_{k}\right)\overset{F[q;u_{k}]\;\mathrm{if}\;x_{k}\;\mathrm{was}\;\mathrm{observed}}{\overset{︷}{\left[q\left(z_{k}|u_{k}\right)\log\frac{q(z_{k}|u_{k})}{p(x_{k},z_{k}|u_{k})}dz_{k}\right]}}dx_{k}}}\;. \end{array}$$

As with the VFE in (Equation 26), we are interested in the minimum of (Equation 30) which once again entails finding $q^{*}$. For clarity of notation we define the solution as
(31)$$\begin{array}{c} G\left(u_{k}\right)=G\left[q^{*};u_{k}\right]=\underset{q}{\arg\;\min}\;G\left[q;u_{k}\right]\;, \end{array}$$
where $G(u_{k})$ is used to compute the policy prior by plugging into (Equation 23). Note that $G(u_{k})$ is a scalar value that denotes the expectation of $F[q^{*};u_{k}]$ under a particular set of constraints on *q* and given a specific action $u_{k}$. To get an intuition for $G(u_{k})$ it can be useful to think of the computation as a two-step procedure consisting of an inner and an outer loop. The inner loop performs variational inference and finds $q^{*}$ conditioned on an action $u_{k}$. The outer loop then computes the resulting EFE by taking the expectation of $F[q^{*};u_{k}]$ under the matching data generating distribution. A core property of EFE is that it introduces an epistemic value term into the optimisation. This leads agents that optimise EFE to seek out areas of state space that have high information gain under the current model, allowing for a principled trade-off between exploration and exploitation [9,28]. To show how this comes about, we can decompose the EFE into a cross-entropy loss and a mutual information (MI) term where the latter quantifies the information gain (in nats or bits) about hidden states $z_{k}$ from observing outcomes $x_{k}$. For the following derivation we will need a bound, the details of which can be found in Section A.3. Starting from (Equation 30), we can factorise the denominator as $p\left(x_{k},z_{k}|u_{k}\right)=p\left(z_{k}|x_{k},u_{k}\right)p\left(x_{k}\right)$, leading to
(32a)$$\begin{array}{cc} G(u_{k})\; & =\;\int \int \;q\left(x_{k}|z_{k}\right)\left[q\left(z_{k}|u_{k}\right)\log\frac{q(z_{k}|u_{k})}{p\left(z_{k}|x_{k},u_{k}\right)p\left(x_{k}\right)}dz_{k}\right]dx_{k} \end{array}$$
(32b)$$\begin{array}{cc} \; & \;\;\;\;\;=\;\int \int \;q\left(x_{k}|z_{k}\right)\left[q\left(z_{k}|u_{k}\right)\left[\log\frac{q(z_{k}|u_{k})}{p(z_{k}|x_{k},u_{k})}-\log p\left(x_{k}\right)\right]dz_{k}\right]dx_{k}\;. \end{array}$$Now we apply the bound from Section A.3 to swap *q* for *p* in the denominator. Making use of this inequality is a standard move across the active inference literature [8,11,13,25,29]. The bound becomes exact when we perform exact inference which is the case in the models we consider here and the discrete models often employed in AIF research, see for instance [8,25]). Instead of applying the bound, another option is to utilise (Equation 30) as is, see [30] for an example. We proceed as
(33)$$\begin{array}{cc} \; & \;\int \int \;q\left(x_{k}|z_{k}\right)\left[q\left(z_{k}|u_{k}\right)\left[\log\frac{q(z_{k}|u_{k})}{p(z_{k}|x_{k},u_{k})}-\log p\left(x_{k}\right)\right]dz_{k}\right]dx_{k}\\ & \;\geq \;\int \int \;q\left(x_{k}|z_{k}\right)\left[q\left(z_{k}|u_{k}\right)\left[\log\frac{q(z_{k}|u_{k})}{q(z_{k}|x_{k},u_{k})}-\log p\left(x_{k}\right)\right]dz_{k}\right]dx_{k}\;. \end{array}$$

Finally we split the integral and integrate over $z_{k}$ to obtain
(34a)$$\begin{array}{cc} \;G(u_{k})\; & \geq \;\int \int \;q\left(x_{k},z_{k}|u_{k}\right)\log\frac{1}{p(x_{k})}dz_{k}dx_{k}-\;\int \int \;q\left(x_{k},z_{k}|u_{k}\right)\log\frac{q(z_{k}|x_{k},u_{k})}{q(z_{k}|u_{k})}dz_{k}dx_{k} \end{array}$$
(34b)$$\begin{array}{cc} \; & =\int q\left(x_{k}|u_{k}\right)\log\frac{1}{p(x_{k})}dx_{k}-\;\int \int \;q\left(x_{k},z_{k}|u_{k}\right)\log\frac{q(z_{k}|x_{k},u_{k})}{q(z_{k}|u_{k})}dz_{k}dx_{k} \end{array}$$
(34c)$$\begin{array}{cc} \; & \;\;=\underset{\text{cross-entropy}}{\underset{︸}{\int q\left(x_{k}|u_{k}\right)\log\frac{1}{p(x_{k})}dx_{k}}}-\underset{\mathrm{Mutual}\;\mathrm{Information}}{\underset{︸}{\;\int \int \;q\left(x_{k},z_{k}|u_{k}\right)\log\frac{q(z_{k},x_{k}|u_{k})}{q\left(z_{k}|u_{k}\right)q\left(x_{k}|u_{k}\right)}dz_{k}dx_{k}}}\;. \end{array}$$In the last line we multiply and divide by $q(x_{k}|u_{k})$ to make the MI term explicit. Readers familiar with the broader AIF literature such as [8,9,12] might not immediately recognise the form of ([Disp-formula FD34c-entropy-23-01565]) as a common decomposition of the EFE. The equivalence between ([Disp-formula FD34c-entropy-23-01565]) and the EFE was originally noted in [8] where the move from ([Disp-formula FD34b-entropy-23-01565]) to ([Disp-formula FD34c-entropy-23-01565]) is done to show the relation between AIF and InfoMax methods. An advantage of writing the EFE as ([Disp-formula FD34c-entropy-23-01565]) is that it cleanly shows how the EFE can be viewed as a combination of two well-known and widely established objectives. From ([Disp-formula FD34c-entropy-23-01565]), we see that $G(u_{k})$ decomposes into a (bound on a) cross-entropy term minus an MI term. Maximising MI is a known way to induce exploration (i.e., information gain about hidden states from observations) in agents and has been employed in multiple settings both within the control theory [31,32] and reinforcement learning literature [17,30,33]. The cross-entropy loss is between a prior $p(x_{k})$ and the posterior distribution $q(x_{k}|u_{k})$ over future observations. This allows for interpreting $p(x_{k})$ as a target/goal prior [25,34]. It endows the agent with an instrumental value term that elicits goal-directed behaviour from inferred policies.

Taking this view, $G(u_{k})$ can be adequately viewed as scoring the behavior resulting from the action $u_{k}$ as a balancing act between MI-based explorative and cross-entropy-based exploitative terms. We now examine each of these terms separately to understand how they work in the linear Gaussian case before considering them jointly. We begin by focusing on the MI and how it may drive exploration when considered in isolation.

### 5.2. Mutual Information Computation

Epistemic behaviour in AIF agents can be considered to be driven by minimising negative MI, as shown in ([Disp-formula FD34c-entropy-23-01565]). MI is in general defined as
(35)$$\begin{array}{cc} I\left[x,z\right]=\;\int \int \;p\left(x,z\right)\log\frac{p(x,z)}{p(x)p(z)}dxdz\; & =H[z]-H[z|x]=H[x]-H[x|z]\;. \end{array}$$

Note that we can write (Equation 35) in terms of entropies of either *x* or *z*. We can do this since MI is symmetric in its arguments. In the LGDS models we consider, we can evaluate the MI component of $G(u_{k})$ as
(36)$$\begin{array}{cc} I[x_{k},z_{k}] & =\frac{1}{2}\log\left|I+\Sigma_{x}^{-1}A\Sigma_{z_{k}}^{-}A^{T}\right|\;. \end{array}$$The detailed derivation of (Equation 36) can be found in Section A.4. To facilitate purely epistemic behaviour, AIF agents can optimise this quantity by selecting appropriate control signals. We will therefore use optimisation of MI as the basis from which to investigate purely epistemic behaviour.

### 5.3. Pure Exploration as a Function of Additive Control Signals

To show the relation between exploration and controls in the additive case, we now need to show how MI depends on the control signal $u_{k}$. For clarity of notation we will do the derivation for the case k>t+1, i.e. we will write in terms of the prior predictive $p\left(z_{k-1}|x_{1:k-2}\right)=N\left(z_{k-1}|\mu_{z_{k-1}}^{-},\Sigma_{z_{k-1}}^{-}\right)$ obtained from (12) instead of $p\left(z_{k-1}|x_{1:k-1}\right)=N\left(z_{k-1}|\mu_{z_{k-1}},\Sigma_{z_{k-1}}\right)$ from (Equation 16). We do so since observations are not available for k>t, meaning we can not perform a full filtering step for the prior time step k−1 unless k=t+1. For the case k=t+1, we can perform filtering for the state prior and can therefore substitute in parameters of $p(z_{k-1}|x_{1:k-1})$ where appropriate. We start the derivation by generating a prediction from our model using (12). We can find the relevant joint distribution at the *k*’th time step by plugging the result into ([Disp-formula FD14b-entropy-23-01565]) to obtain
(37)$$\begin{array}{cc} p\left(\;\left[\begin{array}{c} z_{k}\\ x_{k} \end{array}\right]\;\right)\;\; & =\;N\left(\left[\begin{array}{c} z_{k}\\ x_{k} \end{array}\right]\;|\;\;\left[\begin{array}{c} B\mu_{z_{k-1}}^{-}\;+\;{\hat{\mu }}_{u_{k}}\\ A[B\mu_{z_{k-1}}^{-}\;+\;{\hat{\mu }}_{u_{k}}] \end{array}\right],\;\left[\begin{array}{cc} {\hat{\Sigma }}_{u_{k}}+\Sigma_{z}+B\Sigma_{z_{k-1}}^{-}B^{T} & \left[{\hat{\Sigma }}_{u_{k}}+\Sigma_{z}+B\Sigma_{z_{k-1}}^{-}B^{T}\right]A^{T}\\ A[{\hat{\Sigma }}_{u_{k}}\;+\;\Sigma_{z}\;+\;B\Sigma_{z_{k-1}}^{-}B^{T}] & \;A\left[{\hat{\Sigma }}_{u_{k}}\;+\;\Sigma_{z}\;+\;B\Sigma_{z_{k-1}}^{-}B^{T}\right]A^{T}\;+\;\Sigma_{x} \end{array}\right]\right)\;, \end{array}$$
where we see that the control signal contributes an additive term ${\hat{\Sigma }}_{u_{k}}$ which is the variance associated with the selected action. Interestingly, this means that if we let ${\hat{\Sigma }}_{u_{k}}$ go to 0, the covariance matrix becomes identical to the multiplicative case detailed in Section 5.4. We plug the marginal over states $z_{k}$ into (Equation 36) to get
(38a)$$\begin{array}{cc} I[x_{k},z_{k}] & =\frac{1}{2}\log\left|I+\Sigma_{x}^{-1}A\underset{\Sigma_{z_{k}}}{\underset{︸}{\left[B\Sigma_{z_{k-1}}^{-}B^{T}+\Sigma_{z}+{\hat{\Sigma }}_{u_{k}}\right]}}A^{T}\right| \end{array}$$
(38b)$$\begin{array}{cc} \; & \;\;\;\;\;\;\;\;\;\;=\frac{1}{2}\log\left|I+\underset{\mathrm{Dynamics}\;\mathrm{dependent}}{\underset{︸}{\Sigma_{x}^{-1}AB\Sigma_{z_{k-1}}^{-}B^{T}A^{T}}}+\underset{\mathrm{Policy}\;\mathrm{independent}}{\underset{︸}{\Sigma_{x}^{-1}A\Sigma_{z}A^{T}}}+\underset{\mathrm{Policy}\;\mathrm{dependent}}{\underset{︸}{\Sigma_{x}^{-1}A{\hat{\Sigma }}_{u_{k}}A^{T}}}\right|\;. \end{array}$$We notice that the MI decomposes into three terms. We label the first “Dynamics dependent” since it depends only on the transition matrix *B*, observation noise $\Sigma_{x}$ and the prior state variance $\Sigma_{z_{k-1}}^{-}$. The second term is labeled “Policy independent” since it only depends on the observation noise $\Sigma_{x}$ and transition noise $\Sigma_{z}$. Note that neither of the first two terms are influenced by the control signal. The last term is the only one to include ${\hat{\Sigma }}_{u_{k}}$ and is therefore “Policy dependent”. Crucially, the policy dependent term only depends on the variance of the selected control signal ${\hat{\Sigma }}_{u_{k}}$ and the observation noise $\Sigma_{x}$. In other words, it is *independent* of the latent state $z_{k}$. Since both ${\hat{\Sigma }}_{u_{k}}$ and $\Sigma_{x}$ are available a priori, we can precompute the effect of a policy on the epistemic value term before receiving any observations. Further the result is also independent of the trajectory taken by the agent. Therefore in the case of additive controls, maximising MI does not produce targeted exploration. This necessitates the use of a different model structure when epistemic behaviour is desired. A similar result to ours was obtained by [35] for the case of linear dynamics with additive controls.

### 5.4. Pure Exploration as a Function of Multiplicative Control Signals

To show how epistemic behaviour re-emerges as a function of multiplicative control signals, we now need to show how MI depends on the choice of transition matrix ${\hat{B}}_{k}$. We again proceed by generating a prediction from our model using (11). Plugging this into ([Disp-formula FD14b-entropy-23-01565]) gives us the joint distribution as
(39)$$\begin{array}{cc} p\left(\;\left[\begin{array}{c} z_{k}\\ x_{k} \end{array}\right]\;\right)\;\; & =N\left(\left[\begin{array}{c} z_{k}\\ x_{k} \end{array}\right]|\left[\begin{array}{c} {\hat{B}}_{k}\mu_{z_{k-1}}^{-}\\ A{\hat{B}}_{k}\mu_{z_{k-1}}^{-} \end{array}\right],\left[\begin{array}{cc} \Sigma_{z}+{\hat{B}}_{k}\Sigma_{z_{k-1}}^{-}{\hat{B}}_{k}^{T} & \left[\Sigma_{z}+{\hat{B}}_{k}\Sigma_{z_{k-1}}^{-}{\hat{B}}_{k}^{T}\right]A^{T}\\ A[\Sigma_{z}+{\hat{B}}_{k}\Sigma_{z_{k-1}}^{-}{\hat{B}}_{k}^{T}] & A\left[\Sigma_{z}+{\hat{B}}_{k}\Sigma_{z_{k-1}}^{-}{\hat{B}}_{k}^{T}\right]A^{T}+\Sigma_{x} \end{array}\right]\right)\;. \end{array}$$Plugging the above into (Equation 36) we find that
(40a)$$\begin{array}{cc} I[x_{k},z_{k}] & =\frac{1}{2}\log\left|I+\Sigma_{x}^{-1}A\underset{\Sigma_{z_{k}}}{\underset{︸}{\left[{\hat{B}}_{k}\Sigma_{z_{k-1}}^{-}{\hat{B}}_{k}^{T}+\Sigma_{z}\right]}}A^{T}\right| \end{array}$$
(40b)$$\begin{array}{cc} \; & \;\;\;\;\;\;\;=\frac{1}{2}\log\left|I+\underset{\mathrm{Policy}\;\mathrm{dependent}}{\underset{︸}{\Sigma_{x}^{-1}A{\hat{B}}_{k}\Sigma_{z_{k-1}}^{-}{\hat{B}}_{k}^{T}A^{T}}}+\underset{\mathrm{Policy}\;\mathrm{independent}}{\underset{︸}{\Sigma_{x}^{-1}A\Sigma_{z}A^{T}}}\right|\;\;. \end{array}$$We see that MI now decomposes into two terms. The first term depends on ${\hat{B}}_{k}$ and can be controlled by selecting appropriate transition matrices. The second is *independent* of policy as it only involves process $\Sigma_{z}$ and observation noise $\Sigma_{x}$. Note that similar terms also appear in the additive case. The difference between the additive and multiplicative cases is that the choice of transition matrix ${\hat{B}}_{k}$ is now under the control of the agent. To maximise MI, the agent must therefore select the ${\hat{B}}_{k}$ that maximises the entropy of its latent states $z_{k}$. Taking this view offers a nice intuitive explanation for the resulting exploratory drive: To gain the most information, we must perform the actions that lead to the most uncertain outcomes as described in Section 2. To learn the most, we must sample where we know the least.

### 5.5. Instrumental Value and Expected Free Energy

We now turn our attention to the instrumental value term of $G(u_{k})$ after which we analyse the full EFE construct. Recall from ([Disp-formula FD34c-entropy-23-01565]) that the instrumental value term is a cross-entropy of the form
(41)$$\begin{array}{cc} \int q\left(x_{k}|u_{k}\right)\log\frac{1}{p(x_{k})}dx_{k} & =\int q\left(x_{k}|u_{k}\right)\log\frac{q(x_{k}|u_{k})}{p\left(x_{k}\right)q\left(x_{k}|u_{k}\right)}dx_{k}=KL[q\left(x_{k}|u_{k}\right)||p\left(x_{k}\right)]+H\left[x_{k}|u_{k}\right]\;. \end{array}$$In many cases it is not trivial to obtain $q(x_{k}|u_{k})$ due to intractable integrals. However in the LGDS we are considering, which only involve linear Gaussian relations, it has a tractable expression given by (Equation 17). The KL divergence between two Gaussian distributions is given by
(42)$$\begin{array}{cc} KL[q\left(x_{k}|u_{k}\right)||p\left(x_{k}\right)]\; & =\frac{1}{2}\left(\log\frac{|\Sigma_{p}|}{|\Sigma_{q}|}+n+{\left(\mu_{q}-\mu_{p}\right)}^{T}\Sigma_{p}^{-1}\left(\mu_{q}-\mu_{p}\right)+tr\left[\Sigma_{p}^{-1}\Sigma_{q}\right]\right)\;. \end{array}$$We use subscripts {p,q} to denote whether a term comes from $p(x_{k})$ or $q(x_{k}|u_{k})$ and use {μ,Σ} for the parameters of the corresponding distribution. We can now consider both terms of ([Disp-formula FD34c-entropy-23-01565]) jointly in the case of LGDS. Taking (Equation 35) and (Equation 41) together and making the conditioning on $u_{k}$ in (Equation 35) explicit, we see the full objective comes out as
(43a)$$\begin{array}{cc} \;\;\;\;\;\;\;G(u_{k}) & \geq \underset{\mathrm{Instrumental}\;\mathrm{value}}{\underset{︸}{KL[q\left(x_{k}|u_{k}\right)||p\left(x_{k}\right)]+H\left[x_{k}|u_{k}\right]}}-\underset{\mathrm{Negative}\;\mathrm{MI}}{\underset{︸}{H\left[x_{k}|u_{k}\right]+H\left[x_{k}|z_{k},u_{k}\right]}} \end{array}$$
(43b)$$\begin{array}{cc} \; & =\underset{\mathrm{Risk}}{\underset{︸}{KL[q\left(x_{k}|u_{k}\right)||p\left(x_{k}\right)]}}+\underset{\mathrm{Ambiguity}}{\underset{︸}{H[x_{k}|z_{k},u_{k}]}}\;, \end{array}$$
where we recover the familiar risk and ambiguity terms. In the specific case of LGDS the inequality becomes an equality when we perform exact inference following the equations laid out in Section 4. However note that when combining the instrumental and epistemic terms instead of considering them in isolation, we perform a seemingly innocuous cancellation and remove the entropy $H[x_{k}|u_{k}]$ from the equation. Previously $H[x_{k}|u_{k}]$ appeared twice since we considered the epistemic and instrumental terms separately. However when considering the full EFE construct, this is no longer necessary and we are left with just the ambiguity $H[x_{k}|z_{k},u_{k}]$. Using the entropy expression for a Gaussian distribution, we can write the ambiguity as
(44)$$\begin{array}{cc} H[x_{k}|z_{k},u_{k}] & =\frac{1}{2}\left(n\log2\pi +\log|\Sigma_{22}-\Sigma_{21}\Sigma_{11}^{-1}\Sigma_{12}|+n\right) \end{array}$$Recalling the form of the joint given in ([Disp-formula FD14a-entropy-23-01565]), we can write each block of the covariance matrix out and find
(45a)$$\begin{array}{cc} \;\;\;\;\;\;\;H[x_{k}|z_{k},u_{k}]\; & =\frac{1}{2}(n\log2\pi +\log|\underset{\Sigma_{22}}{\underset{︸}{A\Sigma_{z_{k}}A^{T}+\Sigma_{x}}}-\underset{\Sigma_{21}}{\underset{︸}{A\Sigma_{z_{k}}}}\underset{\Sigma_{11}^{-1}}{\underset{︸}{\Sigma_{z_{k}}^{-1}}}\underset{\Sigma_{12}}{\underset{︸}{\Sigma_{z_{k}}A^{T}}}|+n) \end{array}$$
(45b)$$\begin{array}{cc} \; & \;\;\;\;\;\;\;\;=\frac{1}{2}\left(n\log2\pi +\log|A\Sigma_{z_{k}}A^{T}+\Sigma_{x}-A\Sigma_{z_{k}}A^{T}|+n\right) \end{array}$$
(45c)$$\begin{array}{cc} \; & =\frac{1}{2}\left(n\log2\pi +\log|\Sigma_{x}|+n\right) \end{array}$$The cancellation that follows from using a cross-entropy term to drive goal-directed behaviour means that we are left with only the conditional entropy to drive exploration. The above derivation shows that this term is constant and only depends on the observation noise variance $\Sigma_{x}$. This proves that EFE minimisation in LGDS does not lead to exploration. In fact, minimising a KL divergence between a predicted and desired state (the risk term) is the objective of KL control [36] or message passing based simulations of AIF that minimise variational free energy [6,37]. We conclude that in the case of LGDS, the EFE objective is equivalent to the objective of KL control plus an additive constant that depends only on the observation noise variance.

## 6. Experiments

We investigate the proposed agents in three different settings. First, we investigate pure epistemics in the additive case and show that they do not manifest. Second, we investigate pure epistemics in the multiplicative case and confirm that the agent does indeed perform maximum entropy exploration. Finally we provide comparable experiments for full EFE and show that it indeed reduces to a KL divergence plus a constant.

### 6.1. Pure Epistemics for Additive Controls

In this section we investigate how the epistemic component of EFE behaves in the additive case. In particular we investigate the effects of different transitions on the epistemic value assigned to a policy. For this experiment the transition model is given by (Equation 4). We define the state prior as
(46)$$\begin{array}{cc} p(z_{t-1}|z_{1:t-1}) & =N\left(z_{t-1}|\left[\begin{array}{c} 1\\ 1 \end{array}\right],\left[\begin{array}{cc} 1 & 0\\ 0 & 1 \end{array}\right]\right)\;, \end{array}$$
and set both transition and observation noise to identity matrices. We allow the agent a single action by setting T=t+1, which will be the case for all experiments. Further we define the transition matrix *B*, emission matrix *A* and observation noise $\Sigma_{x}$ as
(47)$$\begin{array}{cc} A=\left[\begin{array}{cc} 1 & 0\\ 0 & 1 \end{array}\right],\;B & =\left[\begin{array}{cc} 1 & 0\\ 0 & 1 \end{array}\right],\;\Sigma_{x}=\left[\begin{array}{cc} 1 & 0\\ 0 & 1 \end{array}\right]\;. \end{array}$$Note that in the additive case, neither matrix has to be time-varying and so we remove the subscripts. We will also use the same parameterization of $\Sigma_{x}$ for all experiments. We compare 4 different candidate parameterisations of the control signal
(48)$$\begin{array}{cc} \Theta_{1}=\{\mu_{1}=\left[\begin{array}{c} 1\\ 1 \end{array}\right],\;\Sigma_{1} & =\left[\begin{array}{cc} 1 & 0\\ 0 & 1 \end{array}\right]\},\;\Theta_{2}=\left\{\mu_{2}=\left[\begin{array}{c} 10\\ 10 \end{array}\right],\;\Sigma_{2}=\left[\begin{array}{cc} 1 & 0\\ 0 & 1 \end{array}\right]\right\}\\ \Theta_{3}=\{\mu_{3}=\left[\begin{array}{c} 1\\ 1 \end{array}\right],\;\Sigma_{3} & =\left[\begin{array}{cc} 3 & 0\\ 0 & 3 \end{array}\right]\},\;\Theta_{4}=\left\{\mu_{4}=\left[\begin{array}{c} 10\\ 10 \end{array}\right],\Sigma_{4}=\left[\begin{array}{cc} 3 & 0\\ 0 & 3 \end{array}\right]\right\}\;. \end{array}$$We choose $\Theta_{1}$ to function as a baseline. For comparison $\Theta_{2}$ shares the same covariance matrix but offers a higher displacement of the mean. $\Theta_{3}$ shares the mean parameter with $\Theta_{1}$ but has higher variance. Finally $\Theta_{4}$ increases both the mean and variance over $\Theta_{1}$. According to (38) varying the mean should not affect the epistemic value since it does not enter into the MI computation. On the other hand, we expect higher variance to affect the policy independent term and lead to increased epistemic value. Consequently we hypothesise that $\Theta_{1}$ and $\Theta_{2}$ will lead to identical results in terms of epistemics even though they result in very different posterior states. Following the same line of reasoning, we hypothesise that $\Theta_{3}$ and $\Theta_{4}$ will lead to identical results. This in turn implies that $\Theta_{1}$ and $\Theta_{3}$ will lead to different values even though the displacement is the same and that a similar pattern will hold for $\Theta_{2}$ and $\Theta_{4}$. Results are shown in Table 1, rounded to 3 digits.

We observe that as hypothesised, MI is not affected by the state transition ($\Theta_{1}$ and $\Theta_{2}$ show identical values). We do find an effect of changing the variance which is again independent of the mean ($\Theta_{3}$ and $\Theta_{4}$ show identical values). This simple experiment confirms our hypotheses given by ([Disp-formula FD38b-entropy-23-01565]): Changing the mean of the control signal does not affect the epistemic term. Changing the variance of the control signal does affect the epistemic term. We conclude that when considering purely epistemic value and additive controls, state transitions and exploration are decoupled. Any effect of the control signal on epistemics is only proportional to the variance of the control, can be pre-computed and does not depend on the agent’s trajectory.

### 6.2. Pure Epistemics for Multiplicative Controls

For comparison, we now perform an analogous experiment for the case of multiplicative controls. We define all quantities in the same way as the additive case. The only change we introduce is defining four transition matrices $B_{1:4}$ to replace $\Theta_{1:4}$. The four candidate transitions we consider are
(49)$$\begin{array}{cc} B_{1} & =\left[\begin{array}{cc} 0.1 & 0\\ 0 & 0.1 \end{array}\right],\;B_{2}=\left[\begin{array}{cc} 1 & 0\\ 0 & 1 \end{array}\right]\;B_{3}=\left[\begin{array}{cc} 10 & 0\\ 0 & 10 \end{array}\right]\;B_{4}=\left[\begin{array}{cc} 100 & 0\\ 0 & 100 \end{array}\right]\;. \end{array}$$Following (40), we hypothesise that larger transitions should lead to lower negative MI by virtue of increasing the value of the policy dependent term. We test this hypothesis across four orders of magnitude and show the results in Table 2.

We observe that, as hypothesised, negative MI decreases as a function of the size of the state transition. Larger transitions lead to lower negative MI though the exact relationship is nonlinear in the size of the transition.

### 6.3. Lack of Epistemics for Expected Free Energy

To investigate the behaviour of AIF agents optimising the full EFE construct, we now repeat both the additive and multiplicative experiments but introduce a goal prior $p(x_{t})$. We define the state prior and the goal as
(50)$$\begin{array}{cc} p(z_{t-1}|x_{1:t-1}) & =N\left(z_{t-1}|\left[\begin{array}{c} 1\\ 1 \end{array}\right],\left[\begin{array}{cc} 1 & 0\\ 0 & 1 \end{array}\right]\right)\;,\;p\left(x_{t}\right)=N\left(x_{t}|\left[\begin{array}{c} 3\\ 3 \end{array}\right],\left[\begin{array}{cc} 3 & 0\\ 0 & 3 \end{array}\right]\right)\;. \end{array}$$Both the multiplicative and the additive agent employ the same emission matrix *A*. For the multiplicative agent we further define the set of candidate transition matrices $B_{1:4}$
(51)$$\begin{array}{cc} A=\left[\begin{array}{cc} 1 & 0\\ 0 & 1 \end{array}\right],\;B_{1} & =\left[\begin{array}{cc} 1 & 0\\ 0 & 1 \end{array}\right],\;B_{2}=\left[\begin{array}{cc} 2 & 0\\ 0 & 2 \end{array}\right],\;B_{3}=\left[\begin{array}{cc} 3 & 0\\ 0 & 3 \end{array}\right],\;B_{4}=\left[\begin{array}{cc} 4 & 0\\ 0 & 4 \end{array}\right]\;. \end{array}$$Here we choose $B_{1}$ as the identity matrix to serve as a baseline. $B_{2}$ moves the agent towards the goal but stops short while $B_{4}$ overshoots by the same amount. This means that either transition puts the agent at the same distance from the goal but with different variances and hence different values of the policy dependent term. Finally we allow $B_{3}$ to move the agent directly to the goal. For the additive case we set the transition matrix $B=B_{1}$ and consider the set of candidate parameterisations $\Theta_{1:4}$
(52)$$\begin{array}{cc} \Theta_{1}=\{\mu_{1}=\left[\begin{array}{c} 0\\ 0 \end{array}\right],\;\Sigma_{1} & =\left[\begin{array}{cc} 1 & 0\\ 0 & 1 \end{array}\right]\},\;\Theta_{2}=\left\{\mu_{2}=\left[\begin{array}{c} 2\\ 2 \end{array}\right],\;\Sigma_{2}=\left[\begin{array}{cc} 1 & 0\\ 0 & 1 \end{array}\right]\right\}\\ \Theta_{3}=\{\mu_{3}=\left[\begin{array}{c} 0\\ 0 \end{array}\right],\;\Sigma_{3} & =\left[\begin{array}{cc} 2 & 0\\ 0 & 2 \end{array}\right]\},\;\Theta_{4}=\left\{\mu_{4}=\left[\begin{array}{c} 2\\ 2 \end{array}\right],\Sigma_{4}=\left[\begin{array}{cc} 2 & 0\\ 0 & 2 \end{array}\right]\right\}\;. \end{array}$$
where we again vary the mean and variance parameters following a similar logic as for the additive experiments. Notably, both $\Theta_{2}$ and $\Theta_{4}$ take the agent directly to the goal but with different variances. We first examine results in the multiplicative case, shown in Table 3.

Here the first column shows the KL divergence between the posterior predictive distribution over observations $q(x_{t}|u_{t})$ and the goal prior $p(x_{t})$ after the corresponding transition. The second column show the additive constant that corresponds to the ambiguity term. The full EFE is displayed in the third column marked *G*. Finally the last two columns display the cross-entropy and negative MI terms as Instrumental and Epistemic value respectively. From Table 3 we see that the lowest KL, and consequently lowest *G*, is obtained when selecting the $B_{2}$ transition matrix. Recall that $B_{2}$ stopped short of the goal while $B_{3}$ placed the agent directly on top of it. However, because controls are multiplicative, $B_{3}$ also results in substantially larger variance which is penalised in the KL. To show that KL is indeed the only driving factor, we can examine the second column, containing the Ambiguity term. We see that it is constant since the observation noise is constant. In turn, we find that the EFE (third column, *G*) can be written as the sum of the KL and Ambiguity columns. For completeness we have also calculated the cross-entropy (Instrumental value) and MI (Epistemic value). Here we observe similar patterns as in the purely exploratory case; larger transitions lead to lower negative MI. This is accurately balanced by the instrumental term though, highlighting an important point: Our result that EFE does not lead to epistemics is only revealed when we consider a particular way of writing the EFE. If we had instead proceeded from the cross-entropy/MI decomposition, the ambiguity constant would not have materialised. We can create a similar table for the additive case, shown in Table 4.

We observe that the lowest KL and *G* corresponds to the transition parameterised by $\Theta_{2}$ as it takes the agent directly to the goal with small variance. What is interesting about Table 4 is the ambiguity column. We obtain the same additive constant as in the multiplicative case which corroborates our results. Even though the dynamics are different and there are substantial differences in both the instrumental and epistemic value terms, the EFE can still be decomposed as a KL and an additive constant that only depends on the observation noise.

## 7. Discussion

Viewing EFE from the point of view of mutual information and cross entropy allows for isolating the epistemic and instrumental value terms so they can be investigated separately. This angle was originally taken in [8] and used as a method of relating AIF to other frameworks. Recent work [1,9] investigates a similar decomposition in the discrete case to highlight how pure exploration and exploitation manifest. Our results as well as [1,8,9] all explore how the EFE operates in specific model architectures. Additionally [1,2,8] also note the equivalence between the mutual information term and the objective of optimal Bayesian design. While work such as [29,30,38] have investigated this link in the general case, deriving the specific equations for a wider class of model architectures promises to be a fruitful area for further research. In those cases, the approach followed in our analysis presents a straightforward way to derive the form of the EFE objective by first decomposing it into a pair of known objective functions and then deriving the expressions separately.

Because EFE can be written in terms of marginal/conditional distributions over the latent states *z*, the analysis presented here applies to any model that utilises a linear likelihood. The results do not depend on the transition model, as demonstrated by our experiments showing similar behaviour for EFE minimization using two different transition models. Our results are consequently equally applicable for a large class of transition models such as auto-regressive models, Gaussian process state space models or deep neural networks without additional adaptation provided the observation model remains linear and Gaussian.

On a similar note, a clear limitation of the present work is the strong reliance on linear observation models. We chose to focus on this case since it allows for an analytical expression of the MI term. However, in general MI is a difficult quantity to compute and one often has to rely on approximations. When approximations are involved, the present analysis is not necessarily applicable, since the decoupling of control signals and epistemics is only demonstrated for the linear case.

In special cases, one can also approximate the joint covariance matrix instead of the mutual information - this is the case for extended Kalman filters for example. In these cases, the present analysis can still apply. Investigating different methods for handling non-linearities is an interesting area for future work on AIF in Gaussian state space models (both linear and non-linear), that can prove useful for neuroscientists and engineers alike.

## 8. Conclusions

In this paper we have shown how to apply AIF in linear Gaussian state space models. We have derived the expressions for EFE in the linear Gaussian case and investigated how the epistemic value terms function. In particular we have shown that in the case of LGDS, EFE reduces to a KL divergence and an additive constant that only depends on observation noise. We therefore conclude that, in the linear Gaussian case, EFE minimization does not lead to epistemic behaviour.

Additionally we have provided an analysis of the epistemic value term considered in isolation, since the cancellation that leads to an absence of epistemic drive for the full EFE is not present when the instrumental term is not included. Our analysis showed that using additive control signals renders the epistemic value term independent of state transitions. This in turn means that any contribution to the epistemic value term is only dependent on the variance associated with the control signal. In other words, it is independent of any observations the agent might receive and any states it may visit, as was previously demonstrated by [35].

Finally we have shown that utilising multiplicative controls, i.e. selecting from a set of candidate transition matrices, circumvents this problem in the purely epistemic case and provides a meaningful interpretation of controls as inducing epistemic behaviour. The resulting setup is reminiscent of the classical Hidden Markov Model that is commonly seen in AIF. Future work can investigate this link by applying recent advances for the discrete case such as [13] to continuous state spaces with multiplicative control signals.

## Figures and Tables

**Figure 1 entropy-23-01565-f001:**
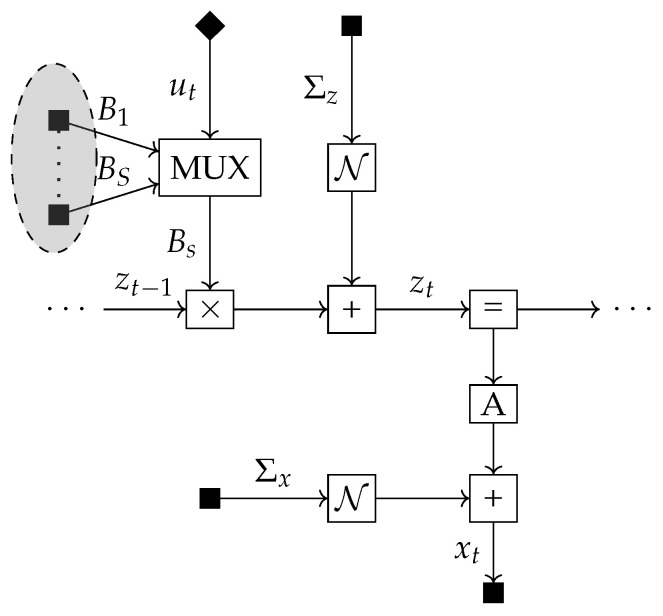
Factor graph of the generative model of an agent with multiplicative control signals.

**Figure 2 entropy-23-01565-f002:**
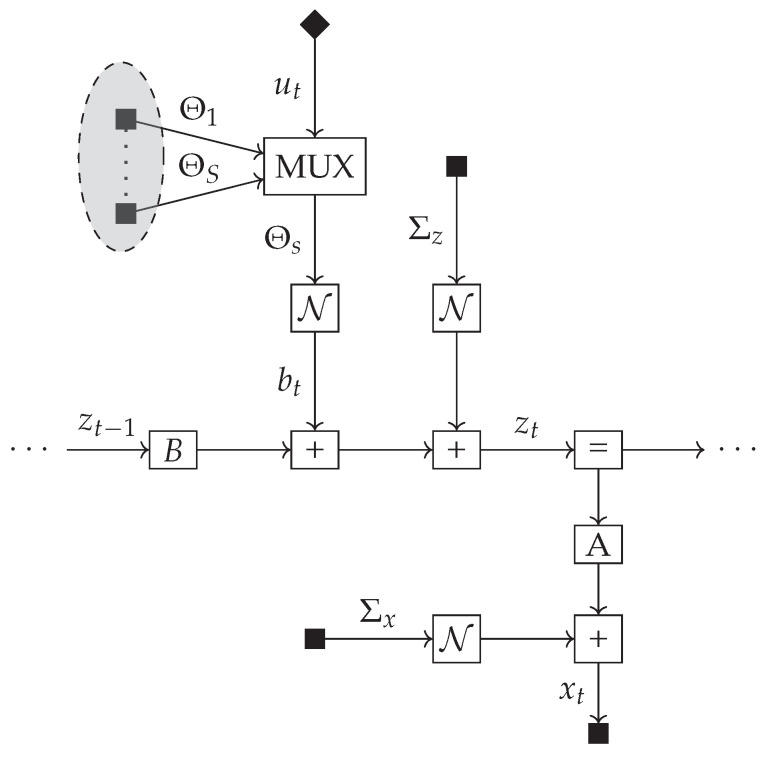
Factor graph of the generative model of an agent with additive controls.

**Table 1 entropy-23-01565-t001:** Epistemic value for additive control signals given state transitions.

Transition	−MI
$\Theta_{1}$	−1.386
$\Theta_{2}$	−1.386
$\Theta_{3}$	−1.609
$\Theta_{4}$	−1.609

**Table 2 entropy-23-01565-t002:** Epistemic value for multiplicative control signals given state transitions.

Transition	−MI
$B_{1}$	−0.698
$B_{2}$	−1.099
$B_{3}$	−4.625
$B_{4}$	−9.211

**Table 3 entropy-23-01565-t003:** EFE for multiplicative controls.

Transition	KL	Ambiguity	G	Instrumental	Epistemic
$B_{1}$	1.33	2.84	4.17	5.27	−1.10
$B_{2}$	0.64	2.84	3.48	5.27	−1.79
$B_{3}$	1.37	2.84	4.21	6.60	−2.40
$B_{4}$	3.54	2.84	6.38	9.27	−2.89

**Table 4 entropy-23-01565-t004:** EFE for additive controls.

Transition	KL	Ambiguity	G	Instrumental	Epistemic
$\Theta_{1}$	3.05	2.84	5.88	7.27	−1.39
$\Theta_{2}$	0.38	2.84	3.22	4.60	−1.39
$\Theta_{3}$	3.16	2.84	5.99	7.60	−1.61
$\Theta_{4}$	0.49	2.84	3.33	4.94	−1.61

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
