# Peer review of "On Epistemics in Expected Free Energy for Linear Gaussian State Space Models"

_entropy, 2021, doi:10.3390/e23121565_

Round 1

Reviewer 1 Report

File uploaded.

Reviewer 2 Report

In this paper, the authors present a declination of the Active Inference (AI) framework on the Linear Gaussian Dynamical Systems (LGDS); they argue that, in the application of AI on LGDS, the epistemic term in the Expected Free Energy (EFE) is ineffective, independently of the specific control chosen for modelling LGDS transitions. Additionally, they analyse the "epistemic power" of the different sorts of control - the additive and the multiplicative one - using a Mutual Information term isolated in EFE, concluding that although the additive control continues to give any epistemic information, multiplicative controls permit state transitions epistemic evaluation. 

The reviewer's opinion is that the spirit of the paper is to encourage: to study applications of AI on artificial/formal systems having engineering importance is of considerable interest. Undoubtedly, the paper is well written and, although it is technical, it is easily readable. The rigour of the mathematical derivations - just a corrigendum in Equation (35) that defines MI without the log function - and the formal approach is justified by the specificity of the result, crucially related to the nature of the distributions involved. On the other hand, the scientific result proposed by the paper has remarkable repercussions on numerous systems whose variables follow Gaussian distributions, and this requires extreme prudence in every formal passage. Probably, in terms of paper organisation, this formal attitude could have been kept in 'Experiments',  where authors have decided to show some numerical cases in the place of more generic - and still manageable - mathematical proofs or graph numerical results as a function of parameters. 

Nevertheless, this reviewer thinks there could be a theoretical bug in the mathematical procedure conditioning the ensuing results. Moving from Eq. (34b) to Eq. (34c), authors consider q(z_k|x_k,u_k) as a factor of the marginalisation of the joint density q(z_k,x_k|u_k) and that provides a term of MI of the variable x_k e z_k in the EFE expression. However, that marginalisation contradicts the crucial assumption made in AI about the form of the joint density q(x_k, z_k|u_k): fixed k, the distribution of hidden states z_k do not depend on observations x_k, i.e., q(x_k, z_k|u_k) = q(z_k|u_k)q(x_k|z_k), where q(x_k|z_k) represent the likelihoods defined in the generative model. Substantially, this position breaks the symmetricity of the joint density and entails a precise causation path where states transit in time and cause observations, not vice versa. Of course, this position alters the remaining derivations: one cannot replace q(z_k|x_k,u_k) with q(z_k,x_k|u_k)/q(x_k,u_k) but using q(x_k, z_k|u_k) = q(z_k|u_k)q(x_k|z_k) in the second term of (34b) one gets the Information Gain term (please, see reference 8 of your bibliography). A straightforward approach would have been to substitute into Equation (30) the generative model of future states p(x_k,z_k|u_k)=q(z_k|x_k,u_k)p(x_k). In both cases, one comes to an EFE expression with a correct "ambiguity" term that, incidentally, cannot be a simple entropy of the outcome distribution but is an expected entropy of the likelihoods over the expected states. That term, or one among its homologous expressions, can be employed to assess the epistemic value related to a series o controls, not MI that is unsuitable to the scope because it evaluates the dependence between variables exclusively.
In summary, the reviewer thinks the authors should provide a deeper discussion about the hypotheses taken to carry out the equations on which their results are based. For this reason, while appreciating the efforts to deepen the behaviour of AI on LGDS, he demands to revise their work in the points discussed before.

Reviewer 3 Report

In this paper, the authors provide a detailed treatment of the expected free energy functional in the case of linear Gaussian state space models. This is very interesting, as a lot of the features of the free energy principle are often mentioned and assumed to be given, but as this paper supports, this is not always apparent. In particular, this paper focuses on the epistemic term, i.e. the information gain on the state belief in the expected free energy.

The paper is well written and offers a nice treatment on Bayesian filtering and expected free energy. A few comments on sections 3 and 4:

  • Section 3, after eq 10 "assuming that the agent selected an action by the procedure described in Section 4". To what extent the procedure by which the action was selected matters, I would think it suffices to say that the selected action is observed?
  • eq 11c, \hat{u}_t : subscript t should go inside the brackets?
  • Section 4, to what extent does it make sense to assume independent control priors, i.e. intuitively I would think that p(u_{t+k}) would also depend on the selected actions on previous steps, i.e. u_{t+1}..u_{t+k-1}?
  • sentence after eq 28 misses end, i.e. I assume "... referring to an arbitrary timestep."
  • "However this expression depends on observations x_k" . Maybe make this explicit in eq 29 by adding x_k in the conditioning of q()?
  • footnote 3, posterior predictive _distribution_ ?

A more fundamental point I would like to raise, and on which I would like to hear the opinion of the authors and see handled better in the manuscript, is the following. The authors use the terms "epistemics" and "exploration" interchangeably in the manuscript. At some point they even state that the expected exploratory behavior would be something like a policy maximizing entropy on the visited states, which is indeed the meaning that it is often attributed in the reinforcement learning context. However, I disagree that this is the meaning of the epistemic term in the expected free energy functional. I would think that actions with a high epistemic value correspond with actions that will probably yield observations that will reduce uncertainty about the beliefs which state one is in. For example if there are different areas in the environment, one area with a lot of "wind", e.g. random perturbations on the dynamics / sensor model, and an other area with less chance of such "wind", you'd expect the agent to prefer to leave the windy area and prefer actions that move it to the other area where there is lower uncertainty on the state.

By instantiating the generative model as a Gaussian state space model as proposed by the authors, I think one by definition assumes there is no uncertainty about the environment to be resolved by action. The only uncertainty about states here is introduced by the characteristics of the actions themselves, as is apparent from the experiments in sections 5.1 and 5.2.

Also in view of KL vs ambiguity, I think indeed by the definition of the generative model, there is no ambiguity that can be resolved by action, as the ambiguity is constant by the definition of the observation noise.

So I think that a) the absense of epistemic behavior in the full FEP formulation is not due to a lacking of features in the FEP, but rather due to the assumptions used to in this particular generative model, and b) that "exploratory behavior" in the sense of maximum entropy in state space, is not to be expected at all from the epistemic term in the expected free energy. I would rather guess that a more "exploratory" behavior might become apparent when also including parameter learning in the loop, which introduces an information gain term to gather samples where the model is not yet accurate, and in this way encourages exploring the whole state space and experience all state space dynamics to learn the correct parameters.

To this end, I also don't think whether the concluding statement "Our results are consequently equally applicable for ... deep neural networks" is valid. State space models with deep neural networks can for example in fact learn (if parameterized as such) varying observation noise depending on the environment context, hence having varying ambiguity. For example, recent work with such models indeed tackles the KL/Ambiguity trade-off [1] and the information gain on state belief [2].

Regarding the experiments section, are all the experiments just evaluating a single future step, or are they working with a longer time horizon / trajectory to roll out? I think an interesting addition would be, if applicable, to have some plots of followed trajectories in a 2D environment for example, to get an intuition of the exploratory vs goal-directed behavior of the agents?

To conclude, I think this paper should definitely be accepted, as the epistemic term in the expected free energy is often not well understood. However, I think the authors should better differentiate between epistemic value as "resolving uncertainty about state belief" and exploration as "maximizing entropy of visited state space", which are to me two distinct things. Also, I think the lacking of epistemic behavior in the experiments is due to the choice of the generative model, rather than due to the FEP formulation per se. I think a more detailed and nuanced discussion of these aspects are required to further improve the paper.

[1] https://doi.org/10.3389/fncom.2020.574372

[2] https://doi.org/10.3389/fnbot.2021.642780

Round 2

Reviewer 1 Report

A separate pdf is uploaded.

Reviewer 2 Report

The authors should declare that their approach departs from the the usual hypothesis - previously mentioned by this reviewer - and discuss it, briefly if they prefer. They should also comment the different interpretations introduced by the infomax vs ambiguity variants of EFE.

Reviewer 3 Report

The authors addressed all my comments and concerns.

Author Response

Thank you for your recommendation and for your time spent reviewing our paper.

Yours sincerely,

Magnus Koudahl